# Application of Prussian Blue in Electrochemical and Optical Sensing of Free Chlorine

**DOI:** 10.3390/s22207768

**Published:** 2022-10-13

**Authors:** Aušra Valiūnienė, Gerda Ziziunaite, Povilas Virbickas

**Affiliations:** Faculty of Chemistry and Geosciences, Vilnius University, Naugarduko 24, LT-03225 Vilnius, Lithuania

**Keywords:** Prussian blue, Prussian white, free chlorine, electrochromic sensor, fluorine-doped tin oxide

## Abstract

In this paper, an electrochemical free chlorine (FCL) sensor was formed by modifying a fluorine-doped tin oxide-coated glass slide (glass|FTO) with a layer of Prussian blue (glass|FTO|PB). The glass|FTO|PB sensor exhibited a wide linear detection range from 1.7 to 99.2 μmol L^−1^ of FCL with a sensitivity of ~0.8 µA cm^−2^ μmol^−1^ L and showed high selectivity for FCL. However, ${\mathrm{ClO}}_{3}^{-}$, ${\mathrm{ClO}}_{4}^{-}$ and ${\mathrm{NO}}_{3}^{-}$ ions have induced only a negligible amperometric response that is highly beneficial for a real-life sample analysis as these ions are commonly found in chlorine-treated water. Moreover, in this work, optical absorption measurement-based investigations of partially reduced PB were carried out as a means to characterize PB catalytic activity towards FCL and to investigate the possibility of applying PB for the optical detection of FCL.

## 1. Introduction

Disinfection is crucial for ensuring the safe consumption of water, since pathogens and microorganisms in untreated water can cause a wide range of diseases [1]. The most common water treatment procedure is chlorination which can be performed using chlorine gas (${\mathrm{Cl}}_{2}$) or hypochlorite (${\mathrm{ClO}}^{-}$) ion-containing compounds, e.g., sodium hypochlorite (NaClO). Dissolution of chlorine gas (Equation (1)) and dissolution of hypochlorite-containing compounds (Equation (2)) results in the formation of free chlorine (FCL), i.e., hypochlorous acid (HClO) and/or a hypochlorite ion (${\mathrm{ClO}}^{-}$) [2,3]. Which form of FCL (HClO or ${\mathrm{ClO}}^{-}$) predominates in disinfected water depends on the alkalinity of the water. In alkaline conditions, equilibrium (Equation (3)) shifts into the formation of a hypochlorite ion, while the formation of hypochlorous acid dominates in acidic conditions [2,3].
(1)$${\mathrm{Cl}}_{2}+H_{2}O\to \;\mathrm{HClO}+\mathrm{HCl}$$
(2)$$\mathrm{NaClO}⇌{\mathrm{Na}}^{+}+{\mathrm{ClO}}^{-}$$
(3)$$\mathrm{HClO}⇌H^{+}+{\mathrm{ClO}}^{-}$$

To ensure the efficiency of water chlorination, as well as to prevent concentrations of FCL exceeding recommended norms (from 0.2 to 1 mg L^−1^ for drinking [4] and from 1 to 5 mg L^−1^ for pool water [5]), it is important to measure the concentration of FCL in chlorine-treated water. There have been analytical methods made for the detection of FCL in water—these include iodometric titration [6], calorimetric detection [7,8,9,10], chromatographic [11,12], chemiluminescence [13,14,15] and green methods [16] While beneficial for some more complex applications, e.g., medical diagnostic procedures performed in vivo in a case of chemiluminescence methods, such analytical methods often require prior synthesis of materials or solutions, involved in the functioning of analytical systems or sophisticated equipment, which prolongs the time required to prepare an analytical system, adds to the price and makes the analytical procedure more complex in general.

Electrochemical sensors are often applied for the detection of FCL [17,18,19] and many other analytes (e.g., fluoroquinolone antibiotics [20,21,22]) because electrochemical detection procedures have high sensitivity and good selectivity, they also often help to reduce problems such as lengthy sample preparation procedures, high analytical costs and the need to use additional reagents [23]. A wide range of electrochemical sensors for FCL detection have been developed. Some of them rely on gold or platinum electrodes, which increases the cost of a sensor [17,18,19]. Platinum electrodes are also known to suffer from passivation due to the formation of platinum oxide [24,25]. Some sensors have been developed using nanoparticles and/or organic molecules to improve the sensitivity or detection limit of the sensors [26,27,28]. The latter is highly desirable since the FCL concentrations used for disinfection can be as low as 0.2 mg L^−1^ of chlorine for drinking water [4] and 1 mg L^−1^ of chlorine for pool water [5]. However, the usage of additional reactive species (nanoparticles and/or organic molecules [26,27,28]) increases the cost, complicates the procedure, generates waste and increases the chance of unwanted side reactions.

Prussian blue (PB) is an inorganic pigment exhibiting unique electrocatalytic properties—PB can participate in redox reactions of some electrochemically active materials (e.g., hydrogen peroxide, ascorbic acid) [29,30,31]; therefore, PB can be applied in the construction of electrochemical sensors and biosensors for various analytes, e.g., glucose [32], ascorbic acid [31], dopamine [33], etc. It is worth mentioning that PB is considered the most advantageous transducer for hydrogen peroxide because the catalytic activity of PB in hydrogen peroxide reduction is ˃1000 higher compared to conventional Pt electrodes [29,30]. Moreover, PB exhibits electrochromic properties, i.e., by switching electrode potential or exposing PB layer to oxidising (or reducing) agents, PB (${\mathrm{KFe}}^{3+}{\left[{\mathrm{Fe}}^{2+}{\left(\mathrm{CN}\right)}_{6}\right]}_{3}$) can be transformed to its colourless reduced form, Prussian white (PW) ($K_{2}{\mathrm{Fe}}^{2+}{\left[{\mathrm{Fe}}^{2+}{\left(\mathrm{CN}\right)}_{6}\right]}_{3}$), or oxidised form, Berlin green (BG) (${\mathrm{Fe}}^{3+}{\left[{\mathrm{Fe}}^{3+}{\left(\mathrm{CN}\right)}_{6}\right]}_{3}$) [34,35,36,37]. This transition between different redox states of PB can be applied in electrochromic sensing [37,38,39,40,41]. For example, the chemical interaction between PB and ascorbic acid has been recently applied for the development of electrochromic self-powered ascorbic acid sensors [42], while the reaction of PW with hydrogen peroxide has been applied for the optical sensing of hydrogen peroxide and glucose [37,38,41,43]. Developing electrochromic sensors and biosensors, especially self-powered ones, seems to be promising as these sensors enable one to save the energy required for analysis [38]. Moreover, the ability to detect analytes (e.g., FCL) by simply observing a change in PB colour could be used for developing new “naked-eye” sensing techniques which allow for the performing of quick and simple qualitative and/or quantitative analysis without using expensive equipment [44].

There are several articles covering the application of Prussian blue (PB) for the electrochemical detection of FCL [45,46,47]. Shim et al. used carbon nanotube/Prussian blue screen-printed composite for the amperometric detection of FCL [45]. This composite showed linear response behaviour covering the range of FCL concentrations being used for the disinfection of drinking water. Salazar et al. worked on PB-based FCL sensors made by covering glassy carbon [46] and screen-printed carbon electrodes [47] with a surfactant-modified PB layer. These amperometric sensors exhibited great sensitivity, worked in wide concentration ranges and appeared to be suitable for FCL detection in water; however, their selectivity was not broadly investigated. Considering that FCL solutions are often polluted with chlorate and perchlorate ions if not stored properly or during the process of manufacturing, the sensitivity towards these species is very important [48].

The objective of this work was to design an amperometric free chlorine (FCL) sensor based on an electrochemically deposited Prussian blue layer on a glass slide covered with fluorine-doped tin oxide (glass|FTO|PB). Considering that the reduced form of Prussian blue, PW, might react with FCL and this reaction should be accompanied by a change in colour; a transparent and electrically conductive glass|FTO electrode was chosen as a substrate for the deposition of PB coating. The amperometric response of the glass|FTO|PB sensor to different concentrations of FCL was investigated and defined by the means of chronoamperometry and cyclic voltammetry. Electrochemical measurements, as well as optical absorption measurement-based experiments allowed for the analysing of FCL-induced oxidation of Prussian white (PW) to PB, as such transitions between oxidised and reduced forms are key principles in which the glass|FTO|PB sensor generates an amperometric response. As a way to inspect the possible application of the developed amperometric FCL sensor for analysis of disinfected water or other real-life samples, the response of the glass|FTO|PB sensor to some potentially interfering compounds (namely ${\mathrm{NO}}_{3}^{-}$, ${\mathrm{ClO}}_{3}^{-}$ and ${\mathrm{ClO}}_{4}^{-}$) was investigated in this research as well.

Although there are several articles that suggest employing PB redox transitions as a means for quantitative FCL analysis [45,46,47], we believe that more in-depth research is lacking before considering the possible practical application of a PB-based FCL sensor in a real-life sample analysis. Therefore, we regard the defining response of the sensor to highly oxidative species, which are often present in FCL solutions as pollutants (${\mathrm{ClO}}_{3}^{-}$ and ${\mathrm{ClO}}_{4}^{-}$), as a novelty. Moreover, to the best of our knowledge, we were the first to conduct optical absorption measurements as a means to characterise PB catalytic activity towards FCL and to investigate the possibility to apply a PB-based sensor for the optical detection of FCL.

## 2. Experimental

### 2.1. Materials

${\mathrm{FeCl}}_{3}\cdot 6H_{2}O$, $K_{3}[\mathrm{Fe}{\left(\mathrm{CN}\right)}_{6}]$, ${\mathrm{KH}}_{2}{\mathrm{PO}}_{4}$, KCl and NaOH of ACS purity and acetone (≥99.8%) were purchased from ROTH (Karlsruhe, Germany). NaClO solution (6–14% active chlorine) was purchased from Emplura (Darmstadt, Germany). ${\mathrm{NaClO}}_{4}$ (≥98%),${\;\mathrm{NaClO}}_{3}$ (≥99%), ${\mathrm{NaNO}}_{3}$ (≥99%), MICRO^®^-90 concentrated cleaning solution, optically transparent and electrically conductive fluorine-doped tin oxide-covered glass slides (glass|FTO) with a thickness of 1 mm were purchased from SIGMA-ALDRICH (Munich, Germany). Water cleaned by Milli Q-plius-Millipore system (Burlington, VT, USA) was used for the preparation of all solutions. Phosphate buffered saline (PBS) used during optical and electrochemical investigations was prepared in 0.2 L volumetric flask by dissolving KCl (0.1 mol L^−1^) and$\;{\mathrm{KH}}_{2}{\mathrm{PO}}_{4}$ (0.01 mol L^−1^) into millipore water. After the dissolution of salts, PBS was alkalized with NaOH up to the pH value of 5.5. The accurate value of pH was determined with the HI83141 analog pH/mV/°C meter equipped with the HI1230B electrode from Hanna Instruments (Bedfordview, Republic of South Africa). Aqueous-free chlorine stock solution of 0.0125 mol L^−1^ concentration was used during all investigations performed in this research. The concentration of free chlorine in a stock solution was checked every day before conducting experiments by applying the iodometric titration procedure [49]. The electrochemically induced reduction in Prussian blue demands the insertion of certain ions ($K^{+}$, ${\mathrm{NH}}_{4}^{+}$, ${\mathrm{Cs}}^{+}$, ${\mathrm{Rb}}^{+}$) which are known to compensate for the excess of negative charges caused by the reduction in ${\mathrm{Fe}}^{3+}$ [37,39,50]. Therefore, to obtain successful redox transitions between PW and PB forms, in this research, all optical and electrochemical investigations of the glass|FTO|PB sensor were performed in $K^{+}$ ion-containing PBS.

### 2.2. Equipment for Electrochemical and Optical Investigations

All electrochemical investigations were carried out using a μAUTOLAB potentiostat/galvanostat from ECOChemie (Utrecht, The Netherlands) in a three-electrode system, consisting of the glass|FTO|PB sensor as the working electrode (geometric surface area 1 cm^2^), Ag|AgCl, KCl_sat._ as the reference electrode (placed as close to the working electrode as possible by using the ‘Luggin capillary’-based connector) and titanium plate as the auxiliary electrode. Electrochemical investigations in FCL-containing PBS were performed in the absence of light considering that FCL degrades under the influence of UV light [51,52]. Optical absorption measurements were carried out using a USB4000 spectrometer from Ocean Optics (Largo, FL, USA).

The surface morphology of the glass|FTO|PB sensor was investigated using a TM 3000 scanning electron microscope (SEM) from Hitachi (Tokyo, Japan).

All investigations were performed at a standard temperature (298 ± 1 K) and standard pressure (101 ± 4 kPa).

### 2.3. Preparation of the Glass|FTO|PB Sensor

The glass|FTO|PB sensor was prepared by electrochemical deposition of the PB layer on the surface of a glass|FTO slide. Prior to deposition of the PB layer, the glass|FTO slide was cleaned by treatment with (i) ultrasound in 2% laboratory dish cleaning solution “Micro90”, (ii) acetone and (iii) deionized water. All three cleaning steps took 16 min each. Then, the PB layer was electrodeposited by applying 40 voltammetric cycles in the potential range of +0.4 to +0.8 V vs. Ag|AgCl, KCl_sat._ (scan rate 40.0 mV s^−1^) [30,37] in the solution of 1 mmol L^−1^ ${\mathrm{FeCl}}_{3}$, 1 mmol L^−1^ $K_{3}\left[\mathrm{Fe}{\left(\mathrm{CN}\right)}_{6}\right]$ and 0.1 mol L^−1^ HCl. Finally, the layer of PB was electrochemically stabilized [30,39,53,54] by applying 20 voltammetric cycles to the glass|FTO|PB electrode in 0.1 mol L^−1^ KCl and 0.1 mol L^−1^ HCl-containing solution in the potential interval from +0.45 to 0 V (vs. Ag|AgCl, KCl_sat._) at a scan rate of 40.0 mV s^−1^.

### 2.4. Preparation of the Glass|FTO|PW and Glass|FTO|PW-PB Electrodes for Optical Investigation

The glass|FTO|PW electrode was prepared via the electrochemical reduction in PB to PW by the means of cyclic voltammetry (CV). It was achieved by applying 20 voltammetric cycles to the glass|FTO|PB sensor in 0.1 mol L^−1^ KCl and 0.1 mol L^−1^ HCl-containing solution in the potential interval from 0 to +0.45 V (vs. Ag|AgCl, KCl_sat._) at a scan rate of 40.0 mV s^−1^, i.e., in the same way as during the stabilization of PB layer, although the cycles were swept in reverse and CV was stopped at the potential of 0 V instead of +0.45 V, as at such potential PB is in its reduced form, PW [29,30,39,55]. Then, the prepared glass|FTO|PW electrode was left in PBS for 3 h while measuring the absorption spectra until the stable optical response was obtained, indicating that a stable PW-PB intermediate was formed [37], thus, resulting in the formation of the glass|FTO|PW-PB electrode.

### 2.5. Electrochemical Investigations of the Glass|FTO|PB Sensor and Optical Investigation of the Glass|FTO|PW-PB Electrode

Cyclic voltammetry-based investigations of the glass|FTO|PB sensor in FCL-containing PBS were performed by sweeping the working electrode (glass|FTO|PB) potential from −0.50 to +0.40 V (vs. Ag|AgCl, KCl_sat._) at a scan rate of 40.0 mV s^−1^.

Chronoamperometric investigations were carried out by applying 0 V (vs. Ag|AgCl, KCl_sat._) potential to the glass|FTO|PB sensor. Aqueous 0.0125 mol L^−1^ stock solutions of free chlorine,${\;\mathrm{ClO}}_{3}^{-}$, ${\mathrm{ClO}}_{4}^{-}$ or ${\mathrm{NO}}_{3}^{-}$ ions were added into PBS to achieve the desired concentration of these species in PBS. Constant stirring at a stirring speed of ~2 revolutions per second (RPS) was used to reduce diffusion limitations and to ensure good mass transport conditions during amperometric measurements. The increments of amperometric signal (*dj*, %) caused by the addition of ${\mathrm{ClO}}_{3}^{-}$, ${\mathrm{ClO}}_{4}^{-}$ and ${\mathrm{NO}}_{3}^{-}$ ions into PBS were compared to FCL-caused increments in the current by using the following Equation (4):(4)$$dj=\frac{\left|j_{i}-j_{0}\right|}{I_{0}}\times 100\%\;$$
where *j*_0_ is the current measured in FCL-containing PBS and *j_i_* is the reduction current measured after addition of${\;\mathrm{ClO}}_{3}^{-}$, ${\mathrm{ClO}}_{4}^{-}$ or ${\mathrm{NO}}_{3}^{-}$ ions of equivalent concentrations into PBS.

Absorption spectroscopy investigation of the glass|FTO|PW-PB electrode was performed in a range of wavelengths from 400 to 1000 nm (VIS-NIR spectrum) in a glass cuvette with a light pathway of 10 mm.

During cyclic voltammetry and absorption spectroscopy investigations, after the addition of FCL stock solution (0.0125 mol L−1) into PBS, FCL-containing PBS was further stirred (~2 RPS) for 1 min and then allowed to settle for 1 min before recording the first signal eliminating any unwanted underwater stream that may interfere with the performance of the measurement.

## 3. Results and Discussion

### 3.1. Scanning Electron Microscopy and Optical Absorption Spectroscopy-Based Investigations of the Glass|FTO|BM Sensor

Prussian blue is a well-known inorganic pigment exhibiting a deep blue colour [56]; thus, a newly formed layer of PB on optically transparent surfaces can be observed by the naked eye. Therefore, the successful deposition of the PB layer on the glass|FTO electrode was indicated by a deep blue colour on the glass|FTO|PB surface. 

In addition, optical absorption spectroscopy investigation of the glass|FTO|PB sensor in PBS indicated that the absorption spectrum of the sensor is typical for PB coatings [43,56,57] with an absorption maximum near 710 nm (Figure 1A). Moreover, in order to ensure the successful formation of the PB layer on the glass|FTO electrode by electrochemical PB deposition in ${\mathrm{FeCl}}_{3}$ and $K_{3}\left[\mathrm{Fe}{\left(\mathrm{CN}\right)}_{6}\right]$ salt-containing solutions, the glass|FTO|PB sensor was investigated by applying scanning electron microscopy (SEM). Elemental SEM analysis (Figure 1B,C) indicated that the glass|FTO|PB sensor contains approx. 2% (by weight) iron confirming the successful deposition of the PB layer on the glass|FTO electrode.

### 3.2. Cyclic Voltammetry-Based Investigation of the Glass|FTO|PB Sensor in FCL-Containing Solution

Considering the strong oxidative properties of FCL and high electrocatalytic activity of Prussian blue [29,30,58,59,60], we expect cyclic voltammograms of the glass|FTO|PB sensor to show a considerable difference in the redox peaks registered in PBS containing various concentrations of FCL. It is depicted in Figure 2A,B that as the concentration of FCL in a buffer solution increases, the cathodic peak current reaches higher values (Figure 2A) and anodic peak current lowers (Figure 2B). This FCL-caused decrement in the anodic current, as well as the increment in cathodic current, could be explained by the oxidation of PW by FCL because less electrons are being subtracted from PW during the electrochemical oxidation of PW into PB and more electrons are needed during the electrochemical reduction in PB into PW if chemical oxidation of PW by FCL occurs simultaneously with the electrochemical redox reactions.

The significant change in the values of peak currents is evident for the anodic scan (Figure 2B) and this change is linearly dependent on the concentration of FCL (Figure 3). Additionally, the cathodic peak current shifts to lower potential values as the concentration of FCL increases corresponding to the same observation made by Salazar et al. in the case of surfactant-modified Prussian blue screen-printed carbon electrode for FCL detection [47], further supporting their claim that such CV tendencies are observed due to PB caused the electrocatalytic reduction in FCL, most likely of a similar nature to the PB-caused electro-catalytic reduction in $H_{2}O_{2}$ [29,30,36,50,61].

Even though the cyclic voltammetry-based procedure (Figure 2) appeared to be suitable to detect FCL by using the glass|FTO|PB sensor in a rather wide linear range from 1.7 to 41.5 μmol L^−1^ (Figure 3) of FCL, the response of this sensor to FCL was found to be non-linear at concentrations higher than 41.5 μmol L^−1^ of FCL (data not presented). This decrement in sensitivity of the glass|FTO|PB sensor towards FCL could be related to the deterioration of PB electrochemical activity due to the repeated cycling of PB between oxidized (PB) and reduced (PW) forms [62]. Therefore, in our further experiments, the response of the glass|FTO|PB sensor to FCL was investigated by the means of chronoamperometry at a constant potential of avoiding cycling-caused degradation of the PB layer.

### 3.3. Chronoamperometric Investigation of the Glass|FTO|PB Sensor in FCL-Containing Solution

Considering that voltammetric data (Figure 2 and Figure 3) indicated a redox reaction between PW and FCL, further investigation of the glass|FTO|PB sensor in FCL-containing PBS was performed by the means of chronoamperometry at a constant potential of 0 V vs. Ag|AgCl, KCl_sat__._ because at this potential, PB exists in its reduced form, PW [29,39,54,55]. As it was expected, the reduction current of the glass|FTO|PB sensor increased after the addition of each amount of FCL into PBS (Figure 4, inset) and was linearly proportionate to the amount of FCL added into the buffer solution (Figure 4). The good linear fit of the data (the coefficient of determination *R*^2^ = 0.997) demonstrates high linearity of the current density (*j*) versus FCL concentration (*c_FCL_*) in a wide concentration range varying from 1.7 to 99.2 μmol L^−1^. This range of concentrations covers and exceeds the amounts of FCL used for both drinking (0.2–1 mg L^−1^ of FCL that translates to 4–19 μmol L^−1^ [4]) and pool water (1–5 mg L^−1^ of FCL that translates to 19–95 μmol L^−1^ [5]) disinfection, suggesting the application of the glass|FTO|PB sensor for both FCL control in disinfected water as well as for analysis of bleaches. The glass|FTO|PB sensor displays a sensitivity of 0.809 µA cm^−2^ μmol^−1^ L, similar to the sensitivity of other PB-based amperometric FCL sensors (Table 1) and appropriate for FCL concentration monitoring in disinfected water.

The glass|FTO|PB sensor appeared to be suitable to measure FCL in PBS chronoaperometrically multiple times (Figure 5). Investigation of the glass|FTO|PB sensor in 50 μmol L^−1^ of FCL-containing PBS indicated that during the first four investigations amperometric response of the glass|FTO|PB sensor to FCL remains constant (~49.9 μA cm^−2^) within the error limits around ±3 μA cm^−2^. Considering the previous investigation [63] indicating that the electrochemical behaviour of the PB layer has been only slightly affected by keeping the PB-coated ITO electrode in a box for 1 year, we believe that our developed glass|FTO|PB sensor for FCL detection could be reused multiple times. Furthermore, it was determined that 1-year-old PB-coated electrode regained its initial electrochemical properties after applying simple cyclic voltammetry-based procedure in acidic KCl solution [63]; thus, PB-covered electrodes create a possibility for the multiple-use sensors that have a potential of decreasing the cost of such electrochemical/electroanalytical devices.

In order to evaluate the usability of the glass|FTO|PB free-chlorine sensor in a real setting, the experiments were carried out in tap water (Figure 6). Considering that (i) investigated tap water was not chlorinated previously and (ii) K+ ions promote PB reduction into PW [37,50], 13 μmol L^−1^ of FCL and 0.1 mol L^−1^ of KCl were added into the tap water sample. As in our previous experiments in this study, the pH value of a sample was adjusted to 5.5 because PB is more stable in acidic media than in alkaline [29]. The value of the current density determined by using the glass|FTO|PB sensor in tap water with 13 μmol L^−1^ of FCL and 0.1 mol L^−1^ of KCl, was equal to −19.93 μA cm^−2^ (Figure 6). This result well-coincides with the current density of −19.97 μA cm^−2^ predicted from the calibration curve (Figure 4) determined for the detection of FCL in PBS. This finding indicates the suitability of the glass|FTO|PB sensor to be used for FCL analysis in chlorinated water. However, in order to apply the glass|FTO|PB sensor for FCL analysis and to obtain correct and reproducible results in real samples (e.g., chlorinated pool or drinking water), the addition of KCl (0.1 μmol L^−1^) and adjusting pH to 5.5 should be performed.

### 3.4. Response of the Glass|FTO|PB Sensor to Interfering Ions

The common issue that occurs, both in the manufacturing and storage of FCL solutions, is pollution with chlorate (${\mathrm{ClO}}_{3}^{-})$ and perchlorate (${\mathrm{ClO}}_{4}^{-})$ ions [48]. The concentration of perchlorate ions tends to increase in time; meanwhile, the amounts of chlorate ions heavily depend on such factors as the concentration of FCL, ionic strength and the presence of transition metal ions [48]. Thus, selectivity towards FCL in the presence of chlorate and perchlorate is of great importance. The decrease in the concentration of FCL in pool water can be impacted by such factors as the degradation of FCL under exposure to UV light [51,52], binding with ammonia or amino groups of organic compounds (formation of combined chlorine, as opposed to free chlorine [2]) and formation of chlorate and perchlorate pollutants. All factors are of interest to be distinguished since the concentration of all the species mentioned must be regulated as their exceeded amounts pose serious health risks [48]. Selectivity towards FCL in the presence of chlorate and perchlorate is hard to achieve, since all these species exhibit strong oxidative properties and display relatively close standard reduction potential values (1.63 V for hypochlorous acid, 1.47 V for chlorate and 1.42 V for perchlorate in acidic conditions [64]). Moreover, nitrate ions are also quite common in groundwater and drinking water [65].

To elucidate the possible impact of these interfering materials (${\mathrm{ClO}}_{3}^{-}$, ${\mathrm{ClO}}_{4}^{-}$ and ${\mathrm{NO}}_{3}^{-}$ ions) on the performance of the sensor we developed, the glass|FTO|PB sensor was investigated in PBS by registering the chronoamperometric response to ${\mathrm{ClO}}_{3}^{-}$, ${\mathrm{ClO}}_{4}^{-}$ and ${\mathrm{NO}}_{3}^{-}$ ions. In Figure 7, we depicted the percentage of change in current (Δ*j*, %) induced by chlorate, perchlorate and nitrate in comparison to FCL-induced reduction current. We demonstrated (Figure 7) that the impact of ${\mathrm{ClO}}_{3}^{-}$, ${\mathrm{ClO}}_{4}^{-}$ and ${\mathrm{NO}}_{3}^{-}$ ions on the reduction current measured with the glass|FTO|PB sensor is negligible; therefore, we are concluding that the glass|FTO|PB sensor has the potential to be applied to the analysis of real-life samples with impurities present in chlorinated water.

### 3.5. Optical Investigation of the Glass|FTO|PW-PB Electrode in FCL-Containing Solution

The electrochromic properties of Prussian blue allow for the characterising of FCL-caused oxidation of PW to PB by the means of absorption spectroscopy. However, PW undergoes spontaneous oxidation to PB in the presence of oxygen; thus, the investigation of impact of FCL on the optical properties of PW requires the avoidance of this oxidation of PW to PB. In our previous research [37], spontaneous oxidation of PW in the presence of oxygen was investigated, revealing the formation of stable intermediate PW-PB form approximately 3 h after the electrochemical reduction in PB to PW. Figure 8 curve 1 depicts the typical absorption spectrum of the glass|FTO|PW-PB electrode obtained by registering optical responses of the glass|FTO|PW electrode until the increase in absorption stops because of the formation of the PW-PB intermediate that does not spontaneously oxidize further. The obtained glass|FTO|PW-PB electrode was investigated by optical spectroscopy to further test the catalytic activity of PW for the reduction in FCL as it was established in the CV investigation (Figure 2). It is seen in Figure 8, curves 2 and 3, that the addition of FCL into buffer solution resulted in an absorption maximum increase. We interpret this result as the oxidation of ${\mathrm{Fe}}^{2+}$ ions to ${\mathrm{Fe}}^{3+}$, therefore, increasing the amount of PB in PW-PB crystal lattice. This finding correlates with the results presented in Figure 2 and Figure 4 confirming the increase in the reduction current as an outcome of the redox reaction between PW and FCL. Therefore, it can be concluded that PW catalyses the reduction in FCL in a similar way as in the case of hydrogen peroxide [37,41].

In order to investigate the possibility to apply the glass|FTO|PW-PB electrode for optical determination of FCL, the kinetics of oxidation of the PW-PB layer was investigated in PBS with different concentrations of FCL (Figure 9). It was determined that the time needed for full oxidation of PW-PB to PB depends on the FCL concentration as the absorption maximum reaches its constant value at 19 min or at 20 min when 60 or 49 µmol L^−1^ of FCL was added into PBS, respectively. However, it is seen in Figure 9 that the values of the absorption maximum significantly differ dependent on the FCL concentration during the first 10–15 min (Figure 9) indicating that the glass|FTO|PW-PB electrode might be applied for quantitative optical detection of FCL, similarly to the application of ITO/PB electrode for optical H_2_O_2_ sensing [43]. It should be noted that more comprehensive studies of the PW-PB oxidation rate of FCL concentration should be performed and these investigations are our future goal.

Nevertheless, the glass|FTO|PW-PB electrode can be applied for the qualitative determination of FCL because affecting the glass|FTO|PW-PB electrode by FCL results in a visually observable change in colour from fair ice-like blue colour of PW-PB to deep blue colour of PB. This visually observable transition in the colour of the glass|FTO|PW-PB electrode can be seen clearly when the concentration of FCL in PBS is higher than 10 µmol L^−1^. In addition, visually observable change in the colour of the glass|FTO|PW-PB electrode takes less than 10 min if the concentration of FCL is higher than 50 µmol L^−1^. Meanwhile, at least 30 min is needed to see a change in colour if the concentration of FCL is smaller than 20 µmol L^−1^. Based on these results we conclude that the glass|FTO|PW-PB electrode developed in this research could be successfully applied for naked-eye detection of free chlorine.

## 4. Conclusions

In the presented study we show for the first time that free chlorine (FCL)-induced oxidation of Prussian white (PW) to Prussian blue (PB) can be used for electrochemical and optical detection of FCL. The main innovation and value of this research is the amperometric response of the glass|FTO|PB sensor to highly oxidative species that are often present in chlorinated water as pollutants (chlorate, perchlorate and nitrate ions). Optical absorption experiments as a means to characterise FCL-caused increment of analytical response as PB electro-catalysed reduction in FCL is the novelty of this research.

In particular, electrochemical investigations of the glass|FTO|PB sensor as well as optical investigation of the glass|FTO|PW-PB electrode showed that the sensitivity of PB towards FCL is based on FCL-caused oxidation of PW, i.e., PB catalyses the reduction in FCL through the PW form. The glass|FTO|PB sensor displayed linear dependency between FCL concentration and the reduction current in a wide concentration range from 1.7 to 99.2 μmol L^−1^ of FCL with a sufficient sensitivity of ~0.8 µA cm^−^^2^ μmol^−1^ L similar to other PB-based amperometric FCL sensors reported [45,46,47] and appropriate for FCL concentration monitoring in disinfected water. The main advantage of the sensor developed in this research is its high selectivity towards FCL as chlorate, perchlorate and nitrate ions induce only negligible amperometric response under the working conditions of the sensor.

The glass|FTO|PW-PB electrode developed in this research could be successfully applied for the naked-eye detection of free chlorine because of the visually observable transition in the colour of the glass|FTO|PW-PB electrode at concentrations of FCL in PBS higher than 10 µmol L^−1^.

Considering that the glass|FTO|PB and glass|FTO|PW-PB-based FCL sensors are easy to prepare and do not require the application of additional reactive species that can potentially interfere with results, it can be concluded that sensors for electrochemical and optical FCL determination presented in this research could be a convenient tool for FCL determination in drinking, pool water and bleaches.

## Figures and Tables

**Figure 1 sensors-22-07768-f001:**
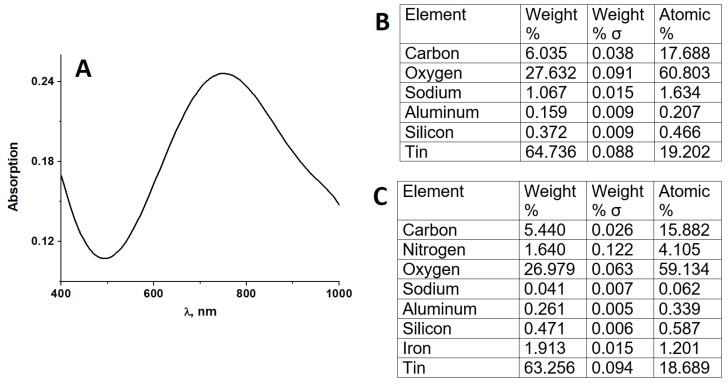
(**A**) optical absorption spectrum of the glass|FTO|PB sensor registered in PBS, (**B**) elemental SEM analysis of the glass|FTO electrode, (**C**) elemental SEM analysis of the glass|FTO|PB sensor.

**Figure 2 sensors-22-07768-f002:**
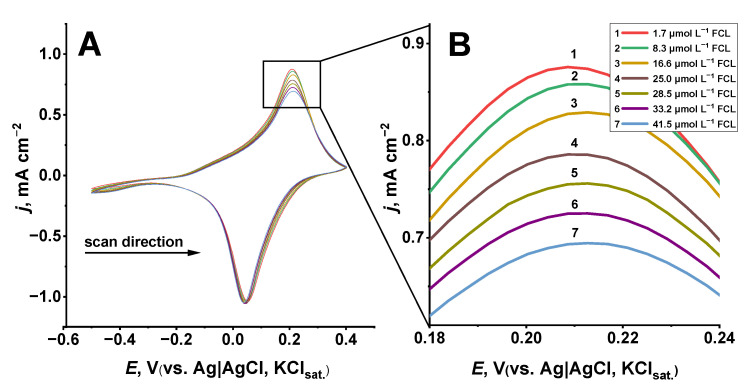
(**A**): CV curves of the glass|FTO|PB sensor registered in PBS containing different concentrations of FCL. (**B)**: magnified data representing anodic current peaks (**A**) of the glass|FTO|PB sensor. Potential scan rate: 40 mV s^−1^.

**Figure 3 sensors-22-07768-f003:**
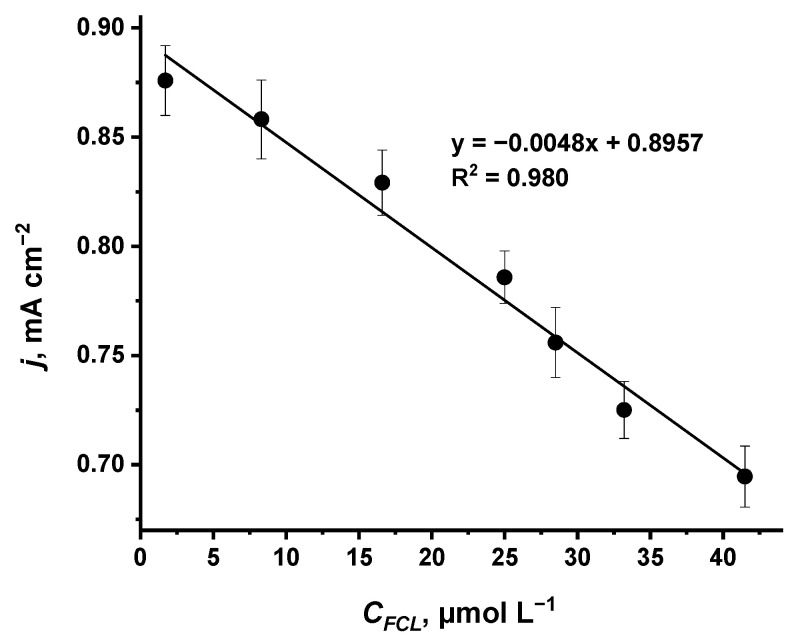
Linear dependency between anodic current peaks (Figure 2) and concentrations of FCL in PBS determined by using the glass|FTO|PB sensor.

**Figure 4 sensors-22-07768-f004:**
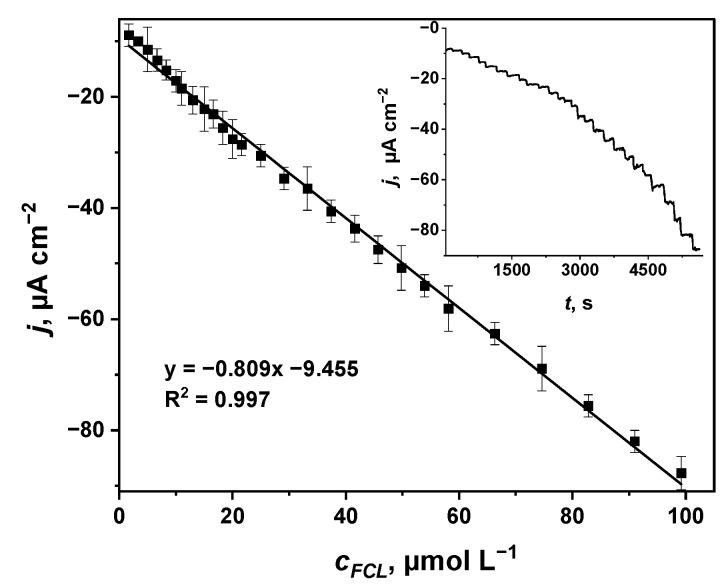
Linear dependence between reduction current of the glass|FTO|PB sensor at 0 V vs. Ag|AgCl, KCl_sat__._ and concentration of FCL ions in PBS. Inset, the chronoamperogram from which cathodic current values for each concentration of FCL in PBS were determined.

**Figure 5 sensors-22-07768-f005:**
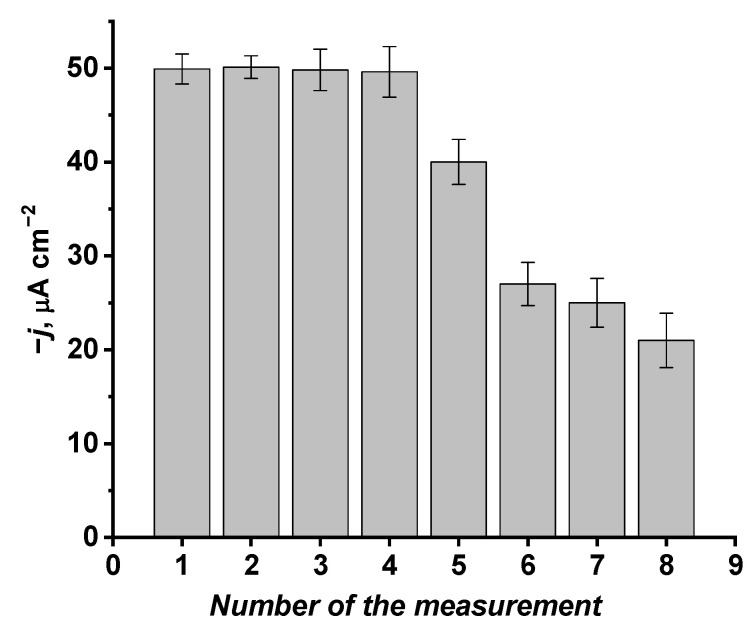
Chronoamperometric investigations performed with the same glass|FTO|PB electrode in 50 μmol L^−1^ of FCL-containing PBS. During chronoamperometric investigations 0 V vs. Ag|AgCl, KCl_sat._ potential was applied to the glass|FTO|PB sensor.

**Figure 6 sensors-22-07768-f006:**
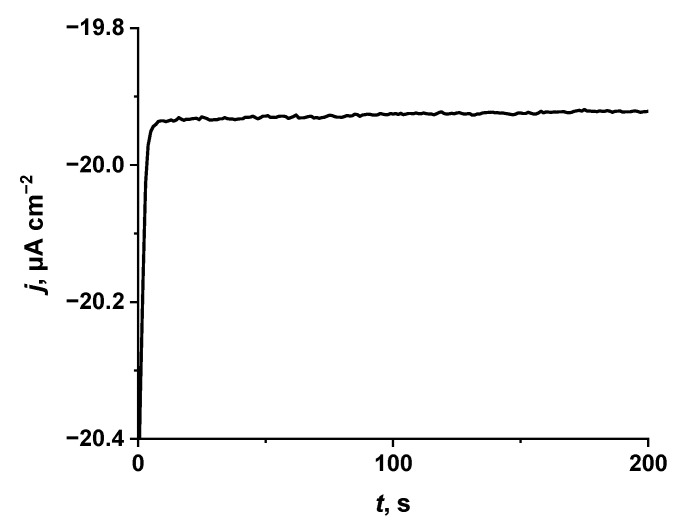
Chronoamperometric investigation of the glass|FTO|PB sensor in tap water (pH 5.5) with 13 μmol L^−1^ of FCL and 0.1 mol L^−1^ of KCl. During chronoamperometric investigation 0 V vs. Ag|AgCl, KCl_sat._ potential was applied to the glass|FTO|PB sensor.

**Figure 7 sensors-22-07768-f007:**
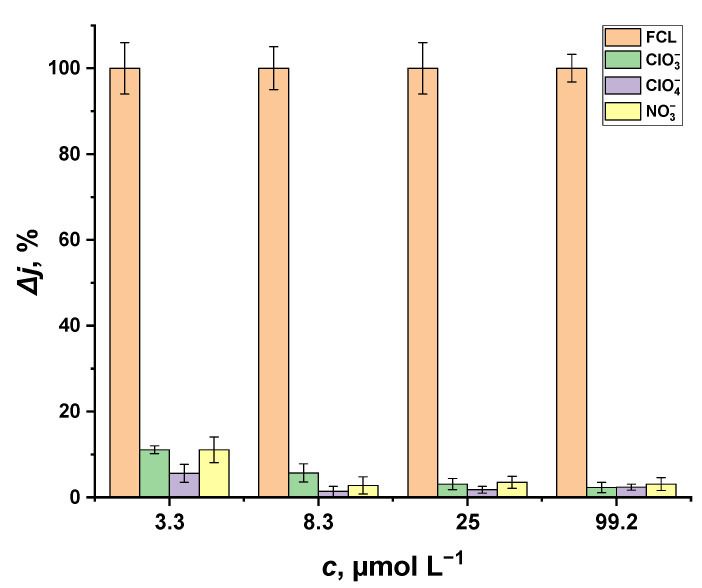
Percentage of change in current (Δ*j*) obtained for the glass|FTO|PB sensor in PBS containing various concentrations of ${\mathrm{ClO}}_{3}^{-}$, ${\mathrm{ClO}}_{4}^{-}$ and ${\mathrm{NO}}_{3}^{-}$ ions. Bias potential: 0 V vs. Ag|AgCl, KClsat.

**Figure 8 sensors-22-07768-f008:**
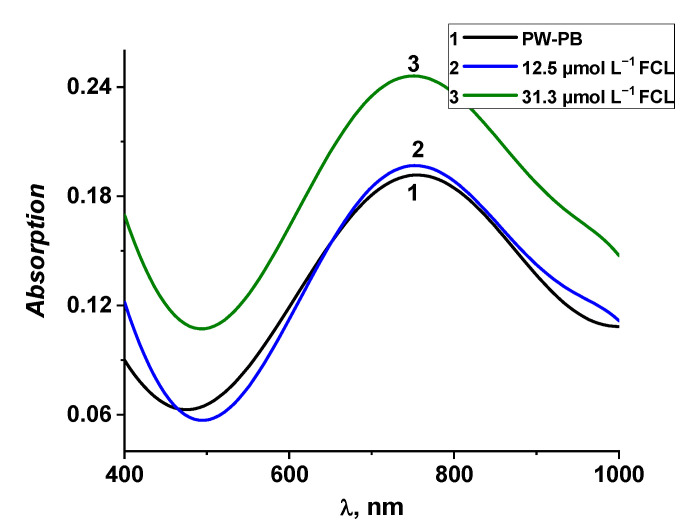
Optical absorption spectra of the glass|FTO|PW-PB electrode registered in PBS without FCL (curve 1); in PBS containing 12.5 μmol L^−1^ of FCL (curve 2); in PBS containing 31.3 μmol L^−1^ of FCL (curve 3). Absorption spectra of the glass|FTO|PW-PB electrode in FCL-containing PBS (curves 2 and 3) were recorded 10 min after the addition of FLC into PBS.

**Figure 9 sensors-22-07768-f009:**
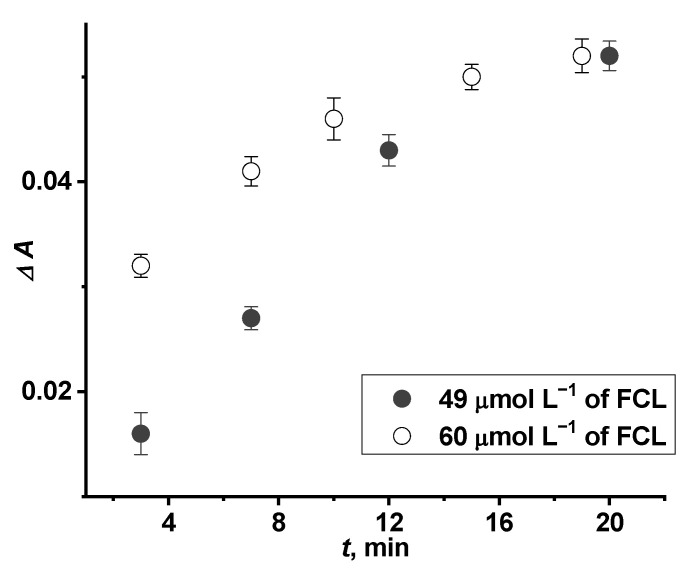
Dependence of change in absorption maximum (Δ*A*) measured after the addition of FCL into PBS on time. Filled circles: Δ*A* measured after adding 49 µmol L^−1^ of FCL into PBS; hollow circles: Δ*A* measured after adding 60 µmol L^−1^ of FCL into PBS.

**Table 1 sensors-22-07768-t001:** Comparison of analytical characteristics of PB-based FCL sensors.

Electrode	Linear Range, µmol L^−1^	Sensitivity
Carbon nanotube/PB paste [45]	1.0–38.1	0.06 µA μmol^−1^ L
Surfactant-modified PB on glassy carbon [46]	0.2–190.6	0.6 µA cm^−2^ μmol^−1^ L
Surfactant-modified PB on screen printed carbon [47]	0.2–57.2	0.9 µA cm^−2^ μmol^−1^ L
Glass|FTO|PB (this work)	1.7–99.2	0.8 µA cm^−2^ μmol^−1^ L

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
