# Peer review of "Application of Prussian Blue in Electrochemical and Optical Sensing of Free Chlorine"

_sensors, 2022, doi:10.3390/s22207768_

Round 1

Reviewer 1 Report

Dear authors,

This is a very nice study on the development of a new hypochlorite sensor based on a Prussian Blue electrode. My recommendation is to accept your manuscript for publication after a minor revision. The work is well structured and very nicely presented, but I have some comments and suggestions that I hope will help improve your work.

Main issues

1 References are missing on Prussian Blue and its applications in sensing. I was surprised not to find any citations to the works of Arkady Karyakin, whose work on Prussian Blue and its analogues is just phenomenal. Please consider citing some of his works, such as:

Karyakin, A. A. (2017). Advances of Prussian blue and its analogues in (bio)sensors. Current Opinion in Electrochemistry5(1), 92–98. https://doi.org/10.1016/j.coelec.2017.07.006   Sitnikova, N. A., Borisova, A. v., Komkova, M. A., & Karyakin, A. A. (2011). Superstable Advanced Hydrogen Peroxide Transducer Based on Transition Metal Hexacyanoferrates. Analytical Chemistry83(6), 2359–2363. https://doi.org/10.1021/ac1033352   The latter, by the way, may give you some clues to improve the stability of your sensors in future works.

2.  the spectroelectrochemical aspects of the work should be highlighted more. One of the things that drew me to referring your work was the use of PB. I am a believer in the use of electrochromic materials for sensing, and I was expecting a little more from your manuscript in this sense. My recommendation is that you try to strengthen that part of the manuscript to demonstrate the usefulness of using PB on a transparent electrode. Otherwise, why not just electrodeposit PB on a conventional graphite screen printed electrode? Makes no sense. My recommendation is for you to discuss a little more on the possibilities of using spectroscopy or image analysis for the determination of FCL, or as a means to assess the status of the sensor over time. Please see my works on this in case they are of help:

Aller Pellitero, M., & del Campo, F. J. (2019). Electrochromic sensors: Innovative devices enabled by spectroelectrochemical methods. Current Opinion in Electrochemistry15, 66–72. https://doi.org/10.1016/j.coelec.2019.03.004

In this kind of situations, it is the rate of color change from PW to PB that will give you the analytical information that you seek. See for instance:

Aller-Pellitero, M., Fremeau, J., Villa, R., Guirado, G., Lakard, B., Hihn, J.-Y., & del Campo, F. J. (2019). Electrochromic biosensors based on screen-printed Prussian Blue electrodes. Sensors and Actuators B: Chemical290(March), 591–597. https://doi.org/10.1016/j.snb.2019.03.100   Zloczewska, A., Celebanska, A., Szot, K., Tomaszewska, D., Opallo, M., & Jönsson-Niedziolka, M. (2014). Self-powered biosensor for ascorbic acid with a Prussian blue electrochromic display. Biosensors and Bioelectronics54, 455–461. https://doi.org/10.1016/j.bios.2013.11.033   Please consider this, and feel free to use the term "electrochromic sensor" in your keywords, or anywhere.   Minor issues

* Please comment on the stability of your sensor. How many measurements can it take? how badly is it affected by sample composition? Looking at its operating conditions, at pH 5.5, can you comment on sample treatment steps? what is the usual pH range in tap water? and in swimming pool water? In other words, what's the usability like for your sensor in a real setting?

* You mention that you use constant stirring "to avoid diffusion induced disturbance". What do you mean? Can you clarify the nature of your stirrer? if it is a magnetic stirrer, how reproducible is it from one experiment to another? If it is another means of stirring, could you describe it so that it is clear that the stirring makes your conditions repetitive?

I hope this was helpful.

Reviewer 2 Report

The authors developed a novel Prussian blue decorated FTO for electrochemical sensing of free chlorine. Although being interesting, I find that there are some major issues with the paper that require addressing prior to this being considered for publication in this journal. I have identified the main points for consideration below:

1.     This manuscript has some spelling typos, style errors and grammatical errors. Pleases carefully check the whole manuscript.

2.     To confirm the successful deposition of Prussian blue, SEM, XRD and XPS of Prussian blue should be provided in the revised manuscript.

3.     The sensing mechanism of the proposed sensor should be clearly illustrated in a scheme.

4.     The applicability of the proposed sensor should be validated by real sample, and the results should be validated by the classic detection techniques.

5.     The electrochemical sensing performance of the proposed sensor should be compared with the previously reported ones.

6.     Error bars should be added in the Figures 2-4.

7.   The advantages of electrochemical sensors should be added in the introduction section. Some recent related references are also recommended to be cited, such as Journal of Hazardous Materials 436 (2022) 129107; Materials Today Chemistry 26 (2022) 101043; TrAC Trends in Analytical Chemistry, 2022, 146, 116487.

Reviewer 3 Report

The manuscript entitled Application of Prussian blue in electrochemical sensing of free chlorine submitted by the group of Authors represents a research on electrochemical detection of free chlorine by sensor made of a a fluorine doped tin oxide coated glass slide with a layer of Prussian blue.

In general, the manuscript is interesting, but it lacks the scientific component and uses some well know relations. The Authors use Prussian blue/white redox transition by free chlorine. The scientific impact is too low and I would not recommend the manuscript for publication.

Round 2

Reviewer 2 Report

The authors have addressed the comments. There is no further comment.

Reviewer 3 Report

The Authors have dome considerable improvements in manuscript formatting. Still, the manuscript uses well known PB and the principle itself is not suitable for such a highly ranked journal.